# Epigenetic Factors Related to Low Back Pain: A Systematic Review of the Current Literature

**DOI:** 10.3390/ijms24031854

**Published:** 2023-01-17

**Authors:** Alberto Ruffilli, Simona Neri, Marco Manzetti, Francesca Barile, Giovanni Viroli, Matteo Traversari, Elisa Assirelli, Fabio Vita, Giuseppe Geraci, Cesare Faldini

**Affiliations:** 1Department of Biomedical and Neuromotor Science—DIBINEM, 1st Orthopaedic and Traumatologic Clinic, IRCCS Istituto Ortopedico Rizzoli, University of Bologna, Via Giulio Cesare Pupilli 1, 40136 Bologna, Italy; 2Medicine and Rheumatology Unit, IRCCS Istituto Ortopedico Rizzoli, Via Giulio Cesare Pupilli 1, 40136 Bologna, Italy

**Keywords:** low back pain, epigenetics, spine, therapeutic target, onset

## Abstract

Low back pain (LBP) is one of the most common causes of pain and disability. At present, treatment and interventions for acute and chronic low back pain often fail to provide sufficient levels of pain relief, and full functional restoration can be challenging. Considering the significant socio-economic burden and risk-to-benefit ratio of medical and surgical intervention in low back pain patients, the identification of reliable biomarkers such as epigenetic factors associated with low back pain could be useful in clinical practice. The aim of this study was to review the available literature regarding the epigenetic factors associated with low back pain. This review was carried out in accordance with Preferential Reporting Items for Systematic Reviews and Meta-Analyses (PRISMA) guidelines. The search was carried out in October 2022. Only peer-reviewed articles were considered for inclusion. Fourteen studies were included and showed promising results in terms of reliable markers. Epigenetic markers for LBP have the potential to significantly modify disease management. Most recent evidence suggests that epigenetics is a more promising field for the identification of factors associated with LBP, offering a rationale for further investigation in this field with the long-term goal of finding epigenetic biomarkers that could constitute biological targets for disease management and treatment.

## 1. Introduction

Low back pain (LBP) is pain referred to the lumbar region of the spine and is one of the most common causes of pain, disability, and social cost in Italy and worldwide [1]. More than 40 ± 20% of the population suffers from LBP at least once in their lifetime; each year, up to 35% of adults experience this symptom, and its prevalence worldwide has increased more than 15% in 10 years [2,3]. The global prevalence of low back pain that limits daily activities was estimated at 7.3% in 2015, accounting for about 540 million people [4].This symptom is a leading cause of global years lived with disability [5], with other musculoskeletal conditions such as arthrosis, neck pain, depressive disorders, and migraine joining it in the top 10, with a lifetime prevalence of >80% [6]. Moreover, it represented the leading cause in 126 of the 195 countries and territories investigated in the GBD 2017 disease and injury incidence and prevalence study [5].

Low back pain includes three distinct sources of pain: axial, radicular, and referred [7]. Axial pain occurs in the vertebral or lumbosacral region, while radicular pain manifests as leg pain with metameric distribution secondary to the irritation of the spinal roots or the posterior ganglion of the spinal roots. Referred pain occurs in a region distant from its source but without metameric distribution [7].

Low back pain can present as acute (when the pain episode ends within 6 weeks), subacute (between 6 and 12 weeks), and chronic (beyond 12 weeks). While most non-chronic patients are acute, with self-limited pain at 6 weeks or less, 10–40% of patients develop symptoms that last longer than 6 weeks [8,9].

Multifactorial causes and risk factors contribute to the pathogenesis of low back pain, classified into two categories: vertebral and extra-vertebral [10] (Figure 1).

Many vertebral pathologies are involved, such as intervertebral disc (IVD) degeneration and herniation, spondyloarthropathies, central and foraminal stenosis, spondylolisthesis, facet arthropathies, sacroiliac joint pain, primary tumors, metastases, infections, and fractures [10,11].

Extra-vertebral low back pain can be caused by urologic [12], vascular [13], gynecologic [14], intestinal [15], prostatic [16], and psychiatric diseases [2].

The medical history is crucial for guiding the diagnostic hypothesis; therefore, the physician must assess all aspects, from family members to the type of work, from psychological behavior to socioeconomic status. He must be well-informed about the work patterns (sitting, standing, bending, lifting weights, driving, etc.); the physical activities practiced (sports, hobbies, etc.); and any underlying medical conditions of the patient. Thus, the physician can direct the diagnosis of low back pain to a vertebral or extra-vertebral cause and proceed with targeted physical examination and imaging.

At present, treatment and interventions for acute and chronic low back pain often fail to provide sufficient levels of pain relief, and full functional restoration can be challenging [2,17].

Barriers to the development of non-opioid pain medications include the lack of validated targets, the paucity of diagnostic markers for pain-related conditions, and the high degree of interindividual variation in response to interventions.

The clinical courses of acute and persistent low back pain typically differ: most cases of acute low back pain recover completely within 4–6 weeks, but persistent low back pain has a poor prognosis, with recovery unlikely.

If pain becomes persistent or chronic, an assessment of pain intensity, associated disability, and the patient’s general condition needs to be carried out. This can be achieved via a physical examination; a detailed patient history; and, eventually, imaging.

Surgical intervention is currently the ultimate solution established for patients with severe chronic low back pain or with conservative treatment failure [17]. It can achieve powerful pain relief but it is characterized by high morbidity and intra- and/or post-operative complications [17].

Like many other conditions, low back pain is influenced by genetic and environmental factors [18,19,20]. Studies of monozygotic and dizygotic twins have evidenced that low back pain has a familial component and that environmental factors are responsible for the variability in pain perception [19]; these include physical and psychological stress, physically demanding work, anxiety, and depression. Moreover, symptoms of anxiety were associated with a higher prevalence of LBP in the total sample analysis [19].

The interaction between the environment and genetics leads to regulatory mechanisms based on gene expression modulation, both via overexpression and gene silencing. Indeed, in eukaryotes, genetic expression is dynamically regulated at the chromatin level by epigenetics, defined as the reversible and heritable changes in gene expression without alterations in the underlying DNA nucleotide sequence.

Epigenetic markers principally include DNA methylation and histone post-transcriptional modifications at specific aminoacidic residues (such as methylation, acetylation, phosphorylation, ubiquitination, and sumoylation) [21]. Epigenetic regulation acts through variations in chromatin accessibility influencing DNA transcription and gene expression [22]. In the vast majority, but not all, cases, DNA methylation corresponds to gene silencing, whereas histone modifications can promote both gene activation or silencing (e.g., lysine acetylation providing transcriptional activation and lysine methylation inducing both activation and repression depending on the histone protein and genomic region).

The role of epigenetics in many pain conditions has been widely described in recent years as a process underlying the development of pathologies such as fibromyalgia [23,24], chronic postoperative pain [25], and low back pain [6].

Considering the significant socio-economic burden and risk-to-benefit ratio of medical and surgical intervention in low back pain patients, the identification of reliable epigenetic factors associated with low back pain could be useful in clinical practice. Epigenetic factors are potential diagnostic, therapeutic, and prognostic tools for predicting the occurrence of low back pain and could therefore be helpful for personalized treatment and disease management. 

The aim of the present systematic review was to check the available English literature concerning epigenetic factors related to low back pain, describe them, and analyze their role as biological targets for disease treatment in this subset of patients.

## 2. Materials and Methods

### 2.1. Review Design

A review of the literature was carried out following the Preferential Reporting Items for Systematic Reviews and Meta-Analyses (PRISMA) guidelines [26].

The Oxford level of evidence scale [27] was used to assess the level of evidence of the included studies. The full version was used to assess randomized and non-randomized clinical trials, whereas the modified version was used to assess all other studies.

Inclusion criteria: papers describing epigenetic factors associated with acute or chronic low back pain published in English peer-reviewed journals. Isolated case reports/series with less than 5 patients, literature reviews, and meta-analyses were excluded. The included articles met the PICO criteria for systematic reviews (Population, Intervention, Comparison, and Outcomes). Different types of studies were considered for inclusion: case series, case–control studies, cohort studies, and genome-wide association studies. These studies were conducted either retrospectively or prospectively.

### 2.2. Search Strategy

Pubmed-MEDLINE, the Cochrane Central Registry of Controlled Trials, Google Scholar, and the Embase Biomedical Database were searched over the years 1990–2022 to identify eligible studies in the English literature describing epigenetic markers related to low back pain. The online literature search was conducted in October 2022 by three reviewers (MM, FB, and GV). The authors stated the following research question: “Are there epigenetic factors correlated with acute or chronic low back pain?”. This research question matched all four PICO concepts. “Acute low back pain”, “chronic low back pain”, “neuropathic pain”, “epigenetic regulation”, “epigenetic variants”, and various alternative terms were considered for each key concept to include the maximum number of articles available in the literature pertaining to the research question. Details on the search strategy are summarized in Appendix A.

The following search items were combined to perform the search: “acute low back pain”, “chronic low back pain”, “neuropathic pain”, “epigenetic regulation”, “epigenetic”, “epigenetic variants”, and “epigenome”.

### 2.3. Study Selection

After screening the titles and abstracts, the full-text articles were obtained and reviewed. A manual search of the bibliography of each of the relevant articles was also performed to identify potentially missed eligible papers. Duplicates were removed. The study selection process carried out in accordance with the PRISMA flowchart is shown in Figure 2. The present systematic review was accepted for registration in the PROSPERO database for systematic reviews [28] (ID: CRD42022360037).

### 2.4. Data Extraction

Two reviewers (MM and SN) extracted the data through a standardized data collection form. Three reviewers (MM, SN, and AR) checked the data for accuracy, and inconsistent results were analyzed for discussion. The extracted data concerning the study design (with the level of evidence), number of patients, demographics of patients, low back pain definition, biological sample, gene/s involved, type of analysis, and results are summarized in Tables 1–3. The following outcomes were considered for analysis: acute or chronic low back pain definition; spine disease causing low back pain; epigenetic factors associated with low back pain and their characteristics; and clinical features of low back pain with various questionnaires such as the “Douleur Neuropathique 4 Questionnaire” (DN4) and the “Short-Form McGill Pain Questionnaire” (SF-MPQ).

### 2.5. Methodological Quality Assessment of Included Studies

The assessment of the methodological quality of the studies was performed using checklist criteria. The quality assessment tool of the National Institutes of Health/National Heart, Lung, and Blood Institute was used [29]. After answering a series of multiple-choice questions, the quality of each study was reported as poor, fair, or good.

## 3. Results

### 3.1. Included Studies

According to the research performed, a total of 14 papers [6,30,31,32,33,34,35,36,37,38,39,40,41] met the inclusion criteria and were considered for review. Of these studies, five [30,31,32,33,34] were retrospective case–control studies and five [6,32,35,36,37] were genome-wide association studies (GWAS). In addition, there were two [38,39] prospective cohort studies, one [40] prospective case–control study and one [41] epigenome-wide association study (EWAS).

According to the Oxford level of evidence scale, five [30,31,32,33,34] studies had a level of evidence of III, while the remaining nine studies had a level of evidence of II [6,32,35,36,37,38,39,40,41].

The studies analyzed both small and large populations (*n* = 10 to 70,633), describing the association between epigenetic factors involved in low back pain.

The included studies were heterogeneous (or lacking data) in spine disease, population demographics, and analysis technique (Table 1, Table 2 and Table 3).

### 3.2. Cohort Characteristics

The studies included in the search reported data on a total of 172,554 individuals who underwent genome sampling and analysis. Considering the studies precisely describing patient demographics, there were 3617 females (65.45%) and 1915 males (34.55%). The median age at genome sampling and analysis ranged from 15.5 ± 1.68 to 58.2 ± 4.4 years. Asiatic populations (Chinese and Japanese) [31,33,34,36,38,40] were the most studied by authors, but Caucasian, Non-Hispanic White and Black, and Anglo-American [6,30,32,35,37,39,40,41,42] populations were also evaluated for possible associations. Two studies did not accurately describe the population demographics [31,37].

### 3.3. Pain and Spine Disease Evaluation

Details of the type of spine disease were reported in six [30,31,33,34,35,36] of the selected studies (Table 1), for a total of 4940 cases of degenerative disc disease (99.5%), 15 of lumbar spine stenosis (0.3%), 4 of lumbar spondylolisthesis (0.1%), and 4 of vertebral fractures (0.1%).

In the other studies, the diagnosis was generic or without accurate subtype distribution (i.e., lumbar disc degeneration or spondylolisthesis) or not reported [6,32,37,38,39,40,41,42].

As for the pain assessment, it was accurately described in seven studies [6,30,37,38,39,40,41] with various questionnaires such as the “Douleur Neuropathique 4 Questionnaire” (DN4) and the “Short-Form McGill Pain Questionnaire” (SF-MPQ). In the other studies, the pain assessment was generic or not reported [31,32,33,34,35,36,42].

### 3.4. Epigenetic Factors Associated with Low Back Pain

In eukaryotes, gene expression is dynamically regulated at the chromatin level by epigenetics, defined as the complex of heritable and reversible changes in gene expression occurring without alterations in the underlying DNA nucleotide sequence [43]. Epigenetic markers principally include DNA methylation (the addition of methyl groups to cytosines within CpG dinucleotides) and histone post-translational modifications (such as methylation; acetylation; phosphorylation; ubiquitination; and the sumoylation of aminoacid residues of H1, H2A, H2B, H3, and H4 histone proteins). These modifications give rise to local chromatin remodeling that, in turn, modifies the accessibility of regulatory elements to genes, thus affecting the transcription rate and gene expression. Regulation by non-coding RNAs such as microRNAs is also part of epigenetics. Epigenetic mechanisms regulate cell differentiation and development and are involved in human disease [44].

**Table 1 ijms-24-01854-t001:** Details of the included studies concerning LBP-related epigenetic regulation: DNA methylation in vertebral tissues. Studies are listed in chronological order.

Study Design (Level of Evidence)	Study Population	Age (Mean/Range) Gender Ethnicity	Pain Assessment/Spine Disease	Type of Biological Sample and Technique Used	Gene(s) Involved	Results	Authors
Retrospective case–control (III)	Low back pain group: 10 Healthy pain-free patients: 23 Non-degenerative IVDs: 5	Low back pain group: 45.6 ± 2.8 - 3 males - 5 females - 2 unknown Healthy pain-free: 41.2 ± 2.3 - 14 females - 9 males Non-degenerative IVDs: 58.2 ± 4.4 5 males Canadian	Scale from 0 to 100 and ODI questionnaire \ Degenerative disc disease	Intervertebral disc DNA bisulphite treatment followed by pyrosequencing	SPARC	SPARC promoter was significantly hypermethylated in patients with low back pain measured with ODI score (*p* < 0.01), in surgical patients (*p* < 0.05), and in lumbar disc degeneration based on MRI image scoring (*p* < 0.001) compared to IVD controls.	M. Tajerian (2011) [30]
Retrospective case–control study (III)	Low back pain group: 14 Control group: 4	NS Chinese	NS \ Control group: vertebral fractures Case: degenerative disc disease	Vertebral Cartilagenous Endplate Quantitative RT-PCR	EZH2	It was found that the expression of EZH2 increased in degenerated human cartilaginous endplates compared to controls (*p* < 0.01). This downregulated SOX9 and upregulated the levels of MMP13 and ADTAMTS4, which resulted in the activation of NF-κb or Wnt/β-catenin signaling.	C. Jiang (2019) [31]
Genome-wide association study (GWAS) (II)	Low back pain spine surgery group: 16 IVD early-stage degeneration: (1) Pfirmann I: 3 (2) Pfirmann II: 3 (3) Pfirmann III: 2 IVD advanced-stage degeneration: (1) Pfirmann IV: 8	Low back pain spine surgery group: 55.6 years Males: 6 (37.5%) Females: 10 (62.5%) Japanese	NS \ Degenerative disc disease: 1 Spinal trauma: 6 Lumbar spine stenosis: 3 Lumbar degenerative spondylolisthesis: 4 Lumbar degenerative scoliosi: 1	Intervertebral disc (nucleous pulposus) bisulfite treatment followed by array-based genome wide methylation analysis	CpGs of the whole genome	A total of 220 differently methylated loci (DML) were identified between early and advanced disc degeneration. Four of these were hypomethylated and 216 hypermethylated in the advanced disc degeneration group.	A. Ikuno (2019) [36]
Retrospective case–control study (III)	Degenerative disc group: 52 Control group: 43	Degenerative disc group: 55.5 ± 3.55 Males: 23 (44%) Females: 29 (56%) Control group: 15.5 ± 1.68 Males: 17 (40%) Females: 26 (60%) Chinese	NS \ Lumbar disc degeneration	Intervertebral disc RNA-seq, RNA scope, and RT-PCR	ALKBH5	The author found an increased expression of ALKBH5 during IVD degeneration and NPC senescence, due to decreased KDM4A-mediated H3K9me3 modification, compared to normal IVD (*p* < 0.05).	G. Li (2022) [34]

**Table 2 ijms-24-01854-t002:** Details of the included studies concerning LBP-related epigenetic regulation: DNA methylation in peripheral blood cells. Studies are listed in chronological order.

Study Design (Level of Evidence)	Study Population	Age (Mean/Range) Gender Ethnicity	Pain Assessment/Spine Disease	Type of Biological Sample and Technique Used	Gene(s) Involved	Results	Authors
Genome-wide association study (GWAS) (II)	4863 individuals from five different study groups with low back pain due to lumbar disc degeneration 38 individuals in one of the cohorts were tested for DNA methylation levels (four monozygotic twin pairs, eight dizygotic twin pairs, and fourteen unrelated indivduals	All individuals: 57.7 years Males: 1605 (33%) Females: 3258 (67%) Caucasian	NS \ Lumbar disc degeneration	Peripheral blood 4863 analyzed in five genotyping studies and collected in one GWAS meta-analysis. Of these, 38 involved analysis for PARK2 methylation by array-based technology.	PARK2	The authors tested for an association between lumbar disc degeneration and DNA methylation variants at three CpG sites in the PARK2 promoter. A significant association between DNA methylation at CpG site cg15832436 and LDD (β = 8.74 × 10^−4^, SE = 2.49 × 10^−4^, *p* = 0.006) was observed. A positive trend was also observed for the other two loci, though without reaching significance.	M.K. Williams (2012) [35]
Prospective cohort study (II)	12 patients suffering from chronic back pain or postherpetic nevralgia	69.3 ± 11.3 (44–81) Males: 5 (41%) Female: 7 (59%) Japanese	Douleur Neuropathique 4 Questionnaire (DN4) or Short-Form McGill Pain Questionnaire (SF-MPQ) \ NS	Peripheral blood whole-blood array-based global DNA methylation analysis (Illumia)	TRPA1	A significant correlation between an increase in the DNA methylation level at the CpG island of the TRPA1 gene and an increase in the DN4 score (*p* = 0.001; r = 0.82), which represents the diversity of the neuropathic pain symptoms. There was also a significant inverse correlation between TRPA1 expression and DN4 (*p* = 0.04; r = −0.65).	N. Sukenaga (2016) [38]
Retrospective case–control study (III)	Chronic low back pain group: 50 Pain-free controls: 48	Chronic low back pain group: 44.5 ± 12.7; (19–85) Males: 22 (44%) Females: 28 (56%) Pain-free controls: 39.9 ± 14.7; (19–85) Males: 25 (52.1%) Females: 23 (47.9%) Non-Hispanic White: 50 Non-Hispanic Black: 48	NS \ NS	Peripheral blood reduced representation bisulfite sequencing	CpGs of the whole genome	The authors identified 28,325 hypermethylated and 36,936 hypomethylated CpG sites (*p* < 0.05). After correcting for multiple testing, the authors identified 159 DMRs (q < 0.01 and methylation difference >10%), the majority of which were in the CpG island (50%) and promoter regions (48%) on the associated genes. The genes associated with the differentially methylated regions were highly enriched in biological processes that have previously been implicated in immune signaling, endochondral ossification, and G-protein-coupled transmissions.	E. N. Aroke (2020) [32]
Genome-wide association study (GWAS) (II)	Discovery cohort: 32 - Control group: 16 - Low back pain group: 16 Validation cohort: 63 - Control group: 16 - Low back pain group: 37	Discovery cohort: - Control females: 43.8 ± 4.6 - Low back pain females: 41.3 ± 3.8 - Control males: 43.8 ± 4.0 - Low back pain males: 42.6 ± 3.6 Males: 16 (50%) Females: 16 (50%) Validation cohort: - Control females: 38.5 ± 3.5 - Low back pain females: 46.1 ± 2.7 - Control males: 43.1 ± 3.2 - Low back pain males: 48.4 ± 2.6 Males: 31 (49%) Females: 32 (51%) Caucasian	Canadian adaptation of NIH low back pain taskforce, DN4 and ODI \ NS	T cells isolated from peripheral blood Array-based methylation analyis (Illumina) after bisulfite treatment, followed by validation by pyrosequencing	850,000 CpG sites	Of the 736,414 CpGs identified in men, 179 were hypermethylated and 240 were hypomethylated in LBP patients compared to controls. Of the 735,863 CpGs identified in women, 601 were hypermethylated and 1895 were hypomethylated (*p*-value < 0.05). The generation of a polygenic methylation score for LBP in men and women with three surrogate CpG loci: cg07420274 for women; cg21149944 and cg22831726 for men. In women, the percentage of methylation at position cg07420274 was 39.5 ± 6 2.7% and 49.7 ± 6 3.2% in the control (*n* = 21) and LBP groups (*n* = 25), respectively (*p* < 0.05). A statistically significant association between methylation at cg07420274 and LBP was observed (OR = 1.05, 95% CI: 1.01–1.11, *p* < 0.03). In men, a statistically significant association was found between LBP and cg21149944 methylation (OR = 0.89, 95% CI: 0.82–0.95, *p* < 0.0015) as well as cg22831726 methylation (OR = 0.9, 95% CI: 0.84–0.96, *p* < 0.0036).	S. Grègoire (2021) [6]
Prospective case–control study (II)	Chronic low back pain group: 15 Acute low back pain: 14 Healthy controls: 16	Chronic low back pain group: 39.4 (8.6) Males: 8 (50%) Females: 8 (50%) Acute low back pain: 33.5 (9.2) Males: 8 (57.1%) Females: 6 (42.9%) Healthy controls: 36.2 (14.3) Males: 6 (37.5%) Females: 10 (62.5%) Black, Asian, White	Brief Pain Inventory (BPI), Short-Form McGill Pain Questionnaire (SF-MPQ), and quantitative sensory testing (QST) \ NS	Peripheral blood ELISA-based DNA methylation and H4 histone acetylation levels quantification	NS	Global histone H4 histone acetylation was higher in participants with pain compared to healthy controls (*p* < 0.05, t = 2.261). The mechanical pain threshold, windup ratio measurement 1, and warm detection threshold at the site of pain were positively correlated with H4 acetylation (all r_p_ = −0.315, *p* < 0.05). Global DNA methylation in cLBP participants was significantly lower than aLBP participants and healthy controls (*p* < 0.05). cLBP participants showed highervL2 mRNA expression than aLBP participants and healthy controls (*p* < 0.05).	C. Eller (2021) [40]
Epigenome-wide association study (EWAS) (II)	Chronic low back pain cohort: 48 Pain-free control cohort: 50	Chronic low back pain cohort: 44.2 ± 12.95 Males: 21 (43.7%) Females: 27 (56.3%) Pain-free control cohort: 39.66 ± 14.51 Males: 26 (52%) Females: 24 (48%) Black, White	NS \ Brief Pain Inventory (BPI), quantitative sensory testing (QST), and measurement of conditioned pain modulation (CPM)	Peripheral blood Array-based DNA methylation analysis	Whole-genome CpGs	Based on CPM efficiency (deficient versus efficient CPM participants), the authors identified 6006 differently methylated CpGs (DMCs) in chronic LBP patients and 18,305 in controls. Most of the DMCs were hypomethylated and annotated to genes of relevance to pain: OPRM1, CACNA2D3, GNA12, LPL, NAXD, and ASHD1 in both groups. Conversely, MAPK-Ras signaling pathways were enriched only in the chronic LBP group (*p* = 0.004).	B.R. Goodin (2022) [45]
Genome-wide association study (GWAS) (II)	Chronic low back pain cohort: 49 - Non-Hispanic Blacks: 25 - Non-Hispanic White: 24 Pain-free control cohort: 49 - Non-hispanic blacks: 24 - Non-Hispanic whites: 25	Chronic low-back pain cohort: - Non-Hispanic Black: 43.5 (10.6) - Non-Hispanic White: 45.8 (14.9) Males: 21 (43%) Females: 28 (57%) Pain-free control cohort: - Non-Hispanic Black: 40.7 (16.5) - Non-Hispanic White: 39.3 (12.6) Males: 25 (51%) Females: 24 (49%) Non-Hispanic Black and White	NS \ NS	Peripheral blood RRBS (reduced representation bisulfite sequencing)	Global DNA methylation	Among participants with chronic low back pain, the authors identified 2873 differently methylated loci (DML) with a difference at least of 10% and *p* < 0.0001, many of those related to pain/nociception processing.	E.N. Aroke (2022) [42]
Genome-wide association study (GWAS) (II)	Discovery cohort: from UK Biobank cohort - Chronic low back pain: 70,633 (304,525 controls) - Acute low back pain: 32,209 (304,525 controls) Replication cohort: from HUNT study - Chronic low back pain: 19,760 - Acute low back pain: 4379 - Controls: 39,983	NS Anglo-American	Pain for more than 3 months, and HUNT2&3 survey \ NS	Peripheral blood DNA genotyping	Global DNA	The authors found 13 genomic loci that reached genome-wide significance in back pain analyses and hypothesized that epigenetic markers would act in brain tissues. The authors found 9 of 13 loci colocalized with epigenetic markers in multiple brain tisseus (*p* < 0.05).	A. Bortsov (2022) [37]

**Table 3 ijms-24-01854-t003:** Details of the included studies concerning LBP-related epigenetic regulation: miRNA regulation. Studies are listed in chronological order.

Study Design (Level of Evidence)	Study Population	Age (Mean/Range) Gender Ethnicity	Pain Assessment/Spine Disease	Type of Biological Sample and Technique Used	Gene(s) Involved	Results	Authors
Prospective cohort study (II)	Chronic low back pain group: 44 (1) Therapy responders: 14 (2) Non-responders: 20 Healthy volunteer group: 20	Chronic low back pain group: 44 (1) Therapy responders: 43 ± 13 Males: 7 (50%) Females: 7 (50%) (2) Non-responders: 47 ± 11 Males: 7 (35%) Females: 13 (65%) Healthy volunteer group: 41 ± 10 Males: 11 (55%) Females: 9 (45%) Caucasian	Pain Numerical Rating Scale (NRS) \ NS	CD4^+^ T cells harvested from peripheral blood Semi-quantitative RNA seq RT PCR	MiRNA-124a MiRNA-150 MiRNA-155	MiRNA-124a (patients: 0.79 ± 0.63 vs. healthy volunteers: 0.30 ± 0.16; *p* < 0.001); miRNA-150 (patients: 0.75 ± 0.21 vs. healthy volunteers: 0.56 ± 0.20; *p* = 0.025); and miRNA-155 (patients: 0.55 ± 0.14 vs. healthy volunteers: 0.38 ± 0.16; *p* = 0.017) were significantly upregulated in CLBP patients when compared with healthy volunteers. After the multidisciplinary treatment program, patients who respond to the treatment showed only an increase in miRNA-124a expression (before treatment: 0.54 ± 0.26 vs. after treatment: 1.05 ± 0.56, *p* = 0.007).	B. Luchting (2016) [39]
Retrospective case–control study (III)	12 subjects affected by degenerative spine disease or thoracolumbar fracture or scoliosis	41.1 (11–68) Males: 4 (33%) Females: 8 (67%) Chinese	NS \ Lumbar disc herniation Lumbar stenosis Thoracolumbar fracture	Intervertebral disc RT-qPCR	FBXO6	RT-qPCR showed that expression of FBXO6 mRNA was significantly higher in nucleus pulposus than in anulus fibrosus tissues (*p* < 0.001). Moreover, FBXO6 was highly expressed in non-degenerated discs and decreased with the severity of degeneration (*p* < 0.001) in relation to miR-133a-5p upregulation, suggesting a role in the mir133a-5p/FBXO6 axis in IVD degeneration. The silencing of FBXO6 cause inhibited proliferation, enhanced apoptosis, suppressed ECM synthesis, and accelerated ECM degradation.	X-F. Du (2021) [33]

So far, the available data concerning the epigenetics of LBP principally pertain to gene expression regulation through DNA methylation or microRNA silencing (Table 3), whereas very few data have been published concerning histone modifications. Most of the available studies focused on the detection of methylated CpGs in selected genomic DNA regions or throughout the genome. We here discriminated the studies investigating tissue-specific epigenetic modifications in vertebral tissues (intervertebral discs and nucleus pulposus tissue, Table 1) from those evaluating peripheral blood epigenetic characteristics (Table 2).

#### 3.4.1. Methylation-Regulated Epigenetic Markers Investigated on Intervertebral Disc Tissue (Table 1)

Tajerian et al. [30], in their retrospective case–control study, investigated the methylation status of the SPARC (Secreted Protein Acydic and Cystein Rich) protein promoter in intervertebral disc DNA from patients with chronic low back pain, controls, and preclinical models. MRI images and ODI scores from patients and pain-free controls were collected to objectively describe the clinical features. SPARC promoter methylation was analyzed by bisulfite mapping in chronic LBP patients and pain-free controls, and higher pain levels and a higher degree of IVD degeneration were found in lumbar MRI compared to controls (*p* < 0.001). Moreover, five out of the thirteen sites of the SPARC promoter had higher levels of methylation in patients compared to controls (*p* < 0.05). SPARC is known to affect collagen fibrillogenesis, bone remodeling, and wound healing [45]. In human IVDs, decreased SPARC expression has been linked to aging and degeneration [46]. Additionally, the targeted deletion of the SPARC gene causes accelerated disc degeneration in old mice and behavioral phenotypes that are similar to chronic LBP in humans. The long-term downregulation of SPARC expression may be crucial in the development of chronic LBP, according to genetic data from mice and clinical observations of its downregulation in humans with IVD degeneration [47,48,49]. This is consistent with decreased protein expression as a function of age and disc degeneration. The study suggests that the age-dependent methylation of the SPARC gene promoter induces SPARC silencing, thus contributing to disc degeneration and low back pain.

In an interesting study, Jiang et al. [31] evidenced the regulatory effect of EZH2 (a histone methyltransferase enhancer) on the expression of SOX9, a cartilage growth and transcriptional factor gene required for chondrogenesis. SOX9 is a disc-degeneration-related gene, preventing chondrocyte hypertrophy and thus inhibiting endochondral ossification, a process seen in IVD degeneration. EZH2’s function is to suppress the expression of various genes, including SOX9, through Histone3 Lysine methylation (H3K27me3). EZH2 inhibition reduces the repressive marker H3K27me3, thus upregulating SOX9 expression and slowing down IVD degeneration, suggesting EZH2 as a possible target to slow down IVD degeneration.

A GWAS analysis of DNA methylation associated with human intervertebral disc degeneration (IVD degeneration) was performed by Ikuno et al. [36] on advanced compared to early degenerated nucleus pulposus tissues obtained from patients undergoing spine surgery for low back pain. They observed different methylomes in the two groups, with the hypermethylation of most loci in advanced degeneration cases, suggesting the involvement of DNA hypermethylation with consequent gene silencing in IVD degeneration. Interestingly, three of the hypermethylated loci (CARD14, EFHD2, and RTNKN2) are involved in the regulation of the NFk-B pathway, known to play a pivotal role in inflammation. The silencing of these gene would contribute to NFk-B activation by inducing the transcription of proinflammatory genes such as TNF-, IL-1, IL-6, and IL-8, as well as disc degeneration by upregulating the expression of matrix-degrading enzymes such as MMPs and ADAMTSs.

Nucleus pulposus cell (NPC) senescence is a critical process for IVD degeneration and, consequently, low back pain. Li et al. [34] observed that the upregulation of ALKBH (a demethylase of N6-methyladenosine in RNA molecules) can induce NPC senescence. Under the epigenetic regulation of histone H3K9me3, ALKBH5-mediated RNA demethylation could induce DNMT3B methyltransferase, which in turn methylates and consequently suppresses the expression of the E4F1 transcription factor, thus contributing to cell senescence through gene silencing. This study interestingly suggests crosstalk between the different levels of methylation regulation (RNA and DNA). It also highlights the therapeutic potential of targeting the DNMT3B/E4F1 axis in treating IVD degeneration.

#### 3.4.2. Methylation-Regulated Epigenetic Markers Investigated on Peripheral Blood (Table 2)

Studies on epigenetic factors regulating LBP performed on peripheral blood cells are mostly genome-wide studies comparing global DNA methylation in patients compared to control groups. These studies frequently highlight different global methylation patterns in patients with the involvement of different cellular pathways or single genes.

Williams et al. [35] reported for the first time a large-scale genome-wide association meta-analysis to identify variants associated with lumbar disc degeneration (LDD) in LBP patients based on a GWA meta-analysis of five Northern European cohorts. They identified a variant in the PARK2 gene associated with LDD. Data were obtained from peripheral blood cell DNA from a subset of 38 individuals (four monozygotic twin pairs, eight dizygotic twin pairs, and fourteen unrelated individuals) investigated for differential DNA methylation levels in the PARK2 (Parkinson Protein 2) promoter. The authors found a positive correlation between cgc15832436 site methylation in the PARK2 promoter and LDD (β = 8.74 × 10^−4^, SE = 2.49 × 10^−4^, *p* = 0.006), suggesting that epigenetic regulation may influence the degeneration of intervertebral discs. PARK2 encodes a protein called parkin, a component of a multiprotein E3 ubiquitin ligase complex mediating the targeting of unwanted proteins for proteasomal degradation. In LDD patients, the hypermethylated PARK2 promoter and the related inhibited PARK2 expression can reduce proteasomal degradation, thus altering the normal cellular environment in intervertebral disc cells, with the increased degradation of intervertebral disc tissue.

The prospective cohort study of chronic pain epigenetics performed by Sukenaga et al. [38] highlighted the importance of TRPA1 (potential ankyrin 1 transient receptor) gene methylation status. The authors harvested peripheral blood samples from 12 LBP patients or postherpetic neuralgia patients and measured their pain status via DN4 (Douleur Neuropathique 4) and the SF-MPQ (Short-Form McGill Pain Questionnaire). After a whole-blood array-based methylation analysis, the authors found a significant correlation between an increase in DNA methylation level at the CpG island of the TRPA1 gene (inducing TRPA1 transcription suppression) and an increase in DN4 scores (*p* = 0.001; r = −0.82), which represent the diversity of neuropathic pain symptoms. The authors also described a significant correlation between a decrease in TRPA1 expression and an increase in DN4 scores (*p* = 0.04; r = −0.65) [50]. Increased TRPA1 promoter methylation and decreased TRPA1 expression in whole blood cells were shown to be related to a reduced heat pain threshold in an investigation of human monozygotic twins [49]. TRPA1 appears to play a pivotal role in the development of chronic pain in humans, and it is included in the functional changes of neuro-immune interactions.

Another study from Grègoire et al. [6] used a genome-wide methylation approach to search for methylation signatures in human T cells. They analyzed the methylation status of 850,000 CpG sites in women and men with chronic low back pain compared to pain-free controls. The authors revealed sex-specific DNA methylation signatures in human T cells discriminating chronic LBP participants from healthy controls. In women, the percentage of methylation at position cg07420274 was 39.5 ± 2.7% vs 49.7 ± 3.2% in the control and low back pain groups, respectively (*p* < 0.05), with a significant association between methylation and low back pain (OR = 1.05, 95% CI: 1.01–1.11, *p* = 0.03). In men, a significant association was found between LBP and cg21149944 methylation (OR = 0.89, 95% CI: 0.82–0.95, *p* = 0.0015), as well as cg22831726 methylation (OR = 0.89, 95% CI: 0.84–0.96, *p* = 0.0036). In conclusion, the authors identified a polygenic DNA methylation sex-specific score from circulating T cells with only three differentially methylated loci, whose methylation allowed the categorization of pain status. Although LBP affects both sexes, these results highlight the striking sex difference in DNA methylation signature, suggesting fundamentally different underlying mechanisms and the possibility of sex-specific epigenetic biomarkers and sex-specific therapeutic approaches.

The results of Gregoire et al. [6] were consistent with those of Dorsey et al.’s study [51], wherein the authors performed a whole-transcriptome analysis, collecting peripheral blood from pain-free individuals, acute LBP patients, and chronic low back pain patients at baseline and at 6 months. The transition from acute to chronic low back pain showed a significant upregulation of mRNAs in the blood coding for genes involved in antigen presentation pathways (MHC class I and II). MHC class II gene upregulation has been associated with other chronic pain conditions including lumbar disc herniation, low back pain, and complex regional pain syndrome.

These results were also consistent with Goodin et al.’s [45] findings, obtained from an epigenome-wide association study (EWAS) on peripheral blood DNA from chronic LBP patients compared to pain-free controls. The study aimed at understanding the differences in the DNA methylation landscape in chronic LBP patients related to efficient or inefficient conditioned pain modulation (CPM). The results suggested the existence of characteristic epigenetic signatures of efficient or inefficient CPM. The authors identified 6006 differently methylated CpG sites in the low back pain cohort, most of them hypomethylated and annotated to genes of relevance for pain such as OPRM1, CACNA2D3, and LPL. New pathways of relevance for pain were enriched only in the chronic LBP group and not in the controls, including MAPK-Ras signaling pathways, suggesting their role in chronic LBP through differential methylation (*p* = 0.004).

By reduced representation bisulfite sequencing (RRBS), Aroke et al. [32] compared the methylation status of non-specific chronic LBP patients to pain-free controls and found 159 differentially methylated regions, enriched in inflammatory pathways and bone maturation, suggesting the role of epigenetics in the pathophysiology of non-specific chronic LBP.

The same group investigated differences in DNA methylation levels between chronic LBP patients of different ethnicities (non-Hispanic White and non-Hispanic Black patients). They identified 2873 differentially methylated loci, many of which were annotated to genes involved in nociception and pain progression (such as Corticotropine, realizing hormone signaling, and the GABA receptor signaling pathway), possibly contributing to the more severe pain and disability observed in the non-Hispanic Black group [42].

Bortsov et al. [37] tried to characterize the molecular and cellular pathways related to chronic versus acute LBP by GWAS and found a substantial genetic contribution to chronic but not acute back pain related to genes expressed in the central nervous system. The authors performed an epigenetic analysis by evaluating SNPs in linkage disequilibrium overlapping with epigenetic features to identify genes and pathways correlated to chronic but not acute LBP heritability.

Eller et al. [40] investigated global DNA methylation and H4 histone acetylation in peripheral blood cells collected from acute and chronic LBP patients at low back pain onset. Participants were also subjected to BPI (Brief Pain Inventory) and SF-MPQ assessments and a quantitative sensory test. DNA methylation levels and H4 acetylation levels were compared to the expression of 84 candidate genes with a possible role in pain onset and modulation. The authors findings showed higher levels of H4 acetylation in participants with LBP compared to controls (*p* < 0.05, t = 2.261). Moreover, H4 acetylation was also positively correlated with somatosensory hypersensitivity. Global DNA methylation levels were lower in chronic compared to acute LBP patients and controls (*p* < 0.05), suggesting the role of hypomethylation in the expression of genes contributing to pain chronicity. In particular, methylation levels were positively correlated with several genes involved in pain control (CX3CR1, GCH1, P2RX, PTGES3, and TNF) and negatively correlated with IL2 expression (lower expression in chronic patients).

#### 3.4.3. Epigenetic Regulation through microRNA Signaling (Table 3)

Some of the investigated LBP and epigenetics studies were dedicated to microRNAs, small noncoding RNAs that participate in the regulation of bone metabolism and osteoclast and osteoblast function [52]. These molecules are epigenetic factors involved in the control of specific molecular pathways in bone-related disorders. MicroRNA activity is expressed through the silencing of gene targets whose mRNA is complementary to the miRNA sequence.

By performing an miRNA expression profile analysis on CD4^+^ T cells harvested from the peripheral blood of chronic LBP patients (divided into therapy responders and non-responders) and healthy volunteers, Luchting et al. [39] described *miRNA124a, miRNA150n*, and *miRNA155* as putative biomarkers of low back pain, since they were significantly upregulated in chronic low back pain patients when compared to healthy volunteers (MiRNA-124a: patients 0.79 ± 0.63 vs. healthy volunteers 0.30 ± 0.16, *p* < 0.001; miRNA-150: patients 0.75 ± 0.21 vs. healthy volunteers 0.56 ± 0.20, *p* = 0.025; and miRNA-155: patients 0.55 ± 0.14 vs. healthy volunteers 0.38 ± 0.16, *p* = 0.017). Moreover, after a multidisciplinary treatment program, patients who responded to the treatment showed only an increase in *miRNA124a* expression (before treatment 0.54 ± 0.26 vs. after treatment 1.05 ± 0.56, *p* = 0.007), suggesting that *miRNA-124a* upregulation is associated with therapy response.

In X. D. Fa et al.’s study [33], the miR-133a-5p/FBXO6 axis was shown to be involved in IVD degeneration, one of the main contributors to LBP. MiR-133a-5p expression aggravates IVD degeneration by targeting and inhibiting FBXO6, a protein highly expressed in healthy discs and progressively downregulated in relation to disc degeneration severity. FBXO6 suppression inhibits cell proliferation, enhances apoptosis, suppresses extracellular matrix synthesis, and accelerates extracellular matrix degradation.

In summary, the hypomethylation of some DNA regions; the hypermethylation of some gene promoters (*SPARC, PARK2*); and the overexpression of some miRNAs (*miR-124a, miR-150, miRNA155,* and *miR133a-5p*) are associated with low back pain and its chronic or acute forms.

## 4. Discussion

Chronic pain syndromes are often associated with epigenetic regulation and rarely with long-term changes in the DNA nucleotide sequence. In fact, many environmental agents, such as early trauma, low socioeconomic status, and depression, are present in the pathophysiology of low back pain and are mediated in part by long-term reprogramming via epigenetic mechanisms [53,54,55].

This recalls the concept of *“nature versus nurture”*, a long-standing debate in biology and society about the balance between two competitive factors that determine one’s *“fate”*: genetics (nature) and environment (nurture) [56].

Understanding epigenetic signatures provides insights into the underlying disease mechanisms and epigenetic markers emerging as suggestive biomarkers for complex diseases, not only elucidating the underlying pathogenesis but also identifying possible therapeutic targets.

For example, literature data have demonstrated broad changes in DNA associated with the progression of scoliotic curves in adolescent idiopathic scoliosis patients [57].

Therapeutic strategies have been engineered against a leading cause of morbidity and mortality worldwide: atherosclerosis [58]. Researchers found that the inhibition of PCSK9 could reduce the risk of coronary heart disease by up to 88%, so they started prescribing patients a small interfering RNA called Inclisiran. This molecule can mimic the body’s natural pathway of RNA interference to specifically prevent proprotein convertase subtilisin/kexin type-9 synthesis, achieving a reduction in LDL of 50% with one shot every 6 months.

Although the role of genetic factors in low back pain development is widely accepted, their role in disease management is still under study.

The control of low back pain is a crucial clinical task, but its epigenetics is still largely unknown. Epigenetic biomarkers could constitute biological targets for disease treatment. The identification of such biomarkers has the potential to improve patient management, minimize unnecessary orthopedic intervention, define the best applicable protocol for orthopedic treatment, and identify the subpopulation of patients in which early surgery could avoid operating on severe-low-back-pain-related spine disease with worse outcomes and more risks. Reliable prognostic factors need to be identified to increase the accuracy of the predictive model, and epigenetic markers might represent ideal candidates for low back pain management.

In the present work, we systematically reviewed the available literature, from 1990 to the present date, concerning epigenetic factors associated with low back pain.

Fourteen papers met the inclusion criteria of the present review, [34,59,60,61,62] identifying two principal epigenetic mechanisms of regulation: DNA methylation and microRNA-mediated gene silencing. In the first group, studies investigating local tissues directly involved in LBP development (intervertebral discs and nucleus pulposus) and studies analyzing peripheral blood cells were included. While data on local tissues are likely to better reflect the molecular and etiological mechanisms of LBP, peripheral blood studies can highlight the predisposing factors and molecules involved in the systemic response to LBP. This dichotomy emphasizes the “two extremes” of low back pain: the core cause and the systemic processing of the pain stimulus. Studies on intervertebral disc tissue have identified epigenetic markers that act as target biomarkers, such as ALKBH, which can promote cellular senescence and contribute to intervertebral disc degeneration [34], or that appear to be responsive to pharmacological modulation, such as EZH2, already used as a target in non-Hodgkin’s lymphoma therapy (Tazemetostat). Peripheral blood studies revealed epigenetic factors that can be used as prognostic markers as well as possible therapeutic targets, such as TRPA1, which has been intensively investigated as a pharmaceutical target for persistent nociceptive pain [59,60], and PARK2, previously known for its heritability in Parkinson’s disease [61].

In addition, the few miRNAs identified as regulators of pathways involved in disc degeneration might also represent target mechanisms for future therapies.

Despite the limited number of available studies and their heterogeneity, available data suggest that epigenetics is a very promising field for the identification of factors associated with low back pain, offering a rationale for further investigation.

To the best of our knowledge, this is the first systematic scoping review to analyze the available evidence pertaining to the epigenetic factors influencing low back pain. This work included an analysis of epigenetic factors, focusing not only on hereditable factors but also on the importance of environmental influences and tissue-related genetic expression in the low back pain phenotype.

The main limitation of the present review was the high heterogeneity among the included studies in terms of study design, values of association, and predictive capacity, possibly representing the principal selection bias of the present work. This heterogeneity could have arisen for different reasons; first of all, the varying LBP etiologies covered in the studies could have played a role. Indeed, spondylolisthesis, vertebral fractures, and vertebral stenosis have different age distributions, gender prevalence, environmental factors, and treatment success rates. This could have generated variability in patient characteristics among the studies on this topic. Moreover, the absence of a clear, internationally recognized definition of low back pain in different populations means that the study conclusions may not be comparable and represents a confounding factor. Another assumption must be made considering that low back pain can be investigated in terms of its cause (i.e., intervertebral herniated disc) or how it is processed in the central nervous system. This dichotomy can cause some variability. Finally, great variability can arise in molecular results depending on the tissue source, DNA preparation and processing methods, and sensitivity of the technique applied; all of these factors are exacerbated by the limited number of papers available, possible methylation modulation due to environmental conditions, and the absence of repeated studies to confirm the most interesting results.

The number of published papers on epigenetic factors related to low back pain is limited, but they are of great interest, even if a final international consensus has not been reached. Defining the factors related to low back pain has the potential to completely revolutionize the clinical management of this common disease.

In the future, comprehensive studies on the epigenetic markers of LBP could be incorporated into a “risk of low back pain scoring system” to predict LBP risk [63]. LBP epigenetic testing is not currently performed in clinical settings, but its potential to guide treatment makes epigenetic characterization an appealing future alternative to pain categorization. For example, Gadd et al. [63] used epigenome-wide data to investigate the correlations between DNA methylation profiles and disease incidence, creating epigenetic scores called Episcores. By testing episcores in an independent cohort, 137 disease-related associations, largely independent from immune cell ratio, lifestyle, and biological aging, were found. Artificial intelligence may be used for this purpose, thanks to the development of algorithms based on deep learning and machine learning, employing data from spine radiographs, patient clinical features, and epigenetic factors to create a complete “tailored” diagnostic tool. Although this approach is fascinating, no clinical studies have attempted it so far.

In the forthcoming years, new biomarkers could be combined with clinical and radiographic parameters, hopefully for the development of new therapeutic strategies based on epigenetic modulators. In line with this mission, further prospective comparative studies with homogeneous architectures and cohorts are needed.

## 5. Conclusions

Epigenetics represents a promising field for the identification of factors associated with low back pain, offering a rationale for further investigation in this field. Molecular tests for low back pain have the potential to significantly modify disease management. This will be achieved only after the identification of reliable markers and an understanding of the underlying biologic pathways. In fact, literature data indicate that epigenetic markers are strongly associated with low back pain, with some of these markers being employed as therapeutic targets in other diseases and therefore also having the potential to be investigated as targets for low back pain therapy.

More data are needed, as well as more studies focused on the tissues involved in the pathology of this condition with prospective designs, homogeneous cohorts of patients in terms of demographic characteristics, uniform sample workflows, and technical assessments.

## Figures and Tables

**Figure 1 ijms-24-01854-f001:**
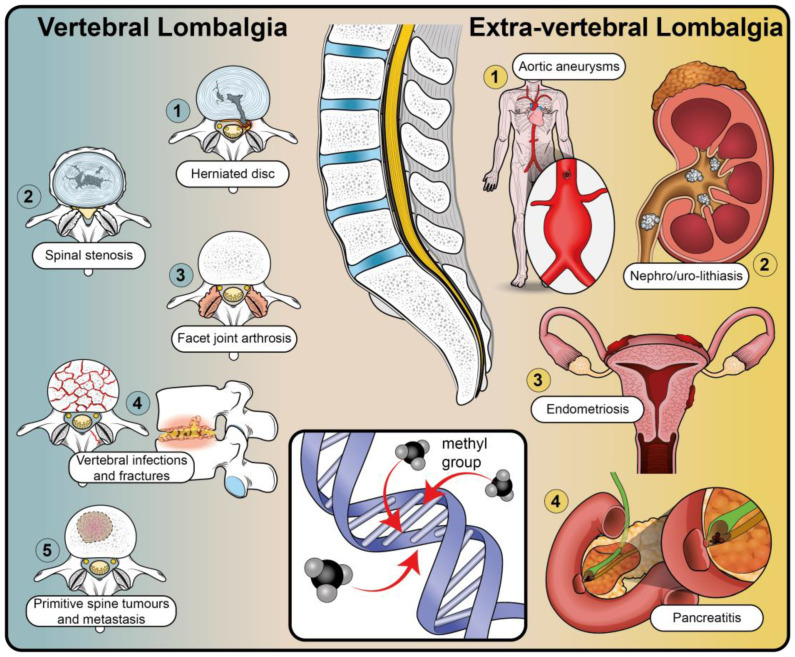
Schematic representation of the principal causes of low back pain: Left—triggers of vertebral lombalgia, the most prevalent being herniated disc, spinal stenosis, and facet joint arthrosis; vertebral fractures are reasonably common in the osteoporotic geriatric and traumatized young populations, whereas vertebral infections and tumors are less common. Centre—epigenetic regulation, represented here by DNA methylation, is one of the possible causes of low back pain. Right—extra-vertebral triggers of low back pain.

**Figure 2 ijms-24-01854-f002:**
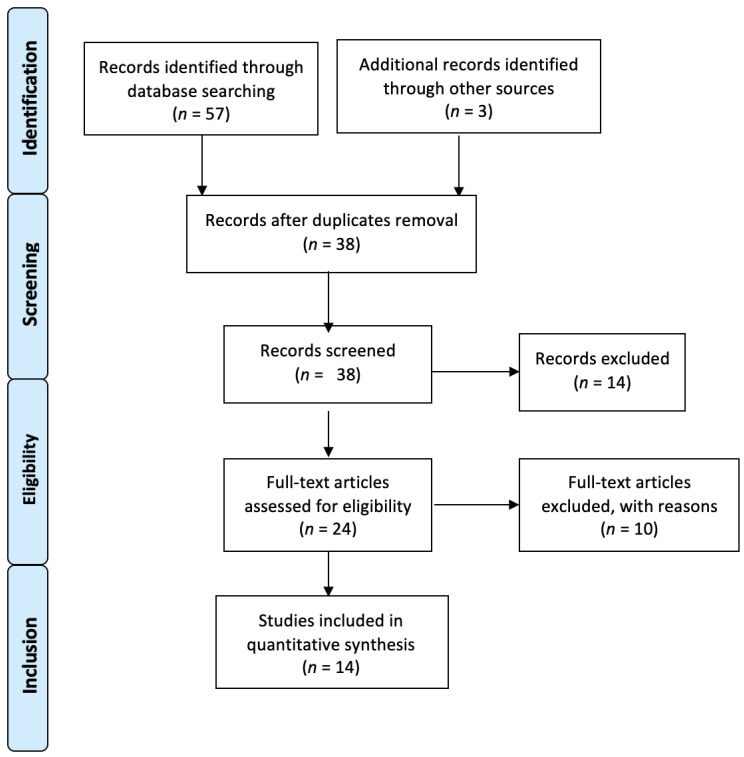
Prisma 2009 flow diagram of the included studies.

## Data Availability

Not applicable.

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
