# Peer review of "Epigenetic Factors Related to Low Back Pain: A Systematic Review of the Current Literature"

_ijms, 2023, doi:10.3390/ijms24031854_

Round 1
Reviewer 1 Report
In this review, the authors try to isolate epigenetic factors associated with low back pain (LBP). For this purpose, they analyzed 14 peer-reviewed studies. As mentioned by the authors, LBP is one of the most common causes of pain, disability and social cost in Italy and worldwide. However, treatment and interventions often fail to provide sufficient pain relief. LBP is a multi-factorial disease with multiple and heterogeneous interactions between environment and genetics. The aim of this study is to find in the literature potential epigenetic factors related to acute or chronic LBP, with a long term goal of finding epigenetic biomarkers that could constitute biological targets for disease treatment.
Although I found this review of potential interest, I found it is lacking from a specific organization trying to extract potential therapeutic application. Indeed, this review is rather descriptive, and is suffering from a lack of critical analysis. The 14 studies filtered by the authors are indeed rather heterogeneous, with a lot of possible confounding effects: age, sex, race origin, cohort analyzed, tissues analyzed, techniques used to study the epigenome. I have the feeling that this a catalog of study cases rather than a review where critical points could be addressed and balanced.
For instance, they based their results on the description of the different papers in the chronological order. Although I understand this could be the easiest way, this is definitely not the best option for such review. I would rather consider to re-organize their analysis by first regrouping some of the studies. For example, I would organize the table into the type of study, i.e. case-control studies, GWAS, retrospective versus prospective studies… Maybe make different tables for the different cases. Then, rather of describing each study one by one, I would rather discuss the different studies as an ensemble and try to confront their similarities (eg. DNA methylation, other histone modifications, miRNA…) and their differences (eg. study design, ethnicity, biological sample…), their advantages and inconvenient, what is the most clear/appealing at the moment…
I would advise the authors to reconsider the organization of this review to be accessible to a broader scientific public. Basically, I am asking here to re-write this review to make a better sens out of it, i.e. make in balance the different approaches/types of study, and try to extract potential therapeutic applications, as originally aimed by the authors.
Other important points to consider:
- I found their main Table 1 very important to summarize the information, but very difficult to read. I would try to fuse some of the columns to make the table easier to read, i.e. Age/Gender/Ethnicity could be fused; Biological sample and Technique could be fused; Authors could be replaced by the reference number… Anything that could compile some of the information and make the table easier to read.
- In their introduction, the authors mentioned DNA methylation and histone post-translational modifications, without mentioning the functional role of these epigenetic modifications on gene expression. At some point, the authors should clearly indicate that hypermethylation (of the promoter) leads to the repression of the genes, histone/H4 acetylation to activation, H3K27/H3K9 trimethylation to repression… This would help the interpretation of some transcriptional up or down-regulation observed in their case studies.
- In the results, it is sometime difficult to follow the function of the modifications, see also my previous comment.
For example, the authors should be more clear on how it works. For instance, SOX9 is a disc degeneration related-gene. Does it mean that he has a positive or a negative effect on the pathology? Inhibition of EZH2 ameliorates the progression of IVDD, which means that inhibition of EZH2 leads to a reduced IVDD progression? In this case, it would be more direct to say: SOX9 is a target of EZH2, the writer of the repressive mark H3K27me3. Inhibition of EZH2 leads to the reduction of this repressive mark, and to an up-regulation of the SOX9 target, which could be implicated in slowing down IVDD progression… For instance, pharmaceutical inhibitors of EZH2 actually do exist (used in cancer treatment), and they could maybe be considered here as potential therapeutic treatment for low back pain relief.
To make this review more attractive, this remark could be extended to some other examples described in this paper.
Author Response
Dear Editor and Reviewers,
First of all, the Authors would like to thank you for the time you dedicated to our manuscript and for your comments and suggestions, which certainly helped us improving the quality of the paper in this second version. It is with great pleasure that we revised our manuscript accordingly. Changes have been made in bold.
Here you can find point-by-point answers to your suggestions and comments. We hope these changes will make the text worth of your consideration.
Best regards,
Simona Neri, M.D.
------------------------------------------------------------------------------------------------------------------------
POINT-BY-POINT ANSWERS:
Reviewer #1:
In this review, the authors try to isolate epigenetic factors associated with low back pain (LBP). For this purpose, they analyzed 14 peer-reviewed studies. As mentioned by the authors, LBP is one of the most common causes of pain, disability, and social cost in Italy and worldwide. However, treatment and interventions often fail to provide sufficient pain relief. LBP is a multi-factorial disease with multiple and heterogeneous interactions between environment and genetics. The aim of this study is to find in the literature potential epigenetic factors related to acute or chronic LBP, with a long-term goal of finding epigenetic biomarkers that could constitute biological targets for disease treatment.
Although I found this review of potential interest, I found it is lacking from a specific organization trying to extract potential therapeutic application. Indeed, this review is rather descriptive, and is suffering from a lack of critical analysis. The 14 studies filtered by the authors are indeed rather heterogeneous, with a lot of possible confounding effects: age, sex, race origin, cohort analyzed, tissues analyzed, techniques used to study the epigenome. I have the feeling that this a catalog of study cases rather than a review where critical points could be addressed and balanced.
For instance, they based their results on the description of the different papers in the chronological order. Although I understand this could be the easiest way, this is definitely not the best option for such review. I would rather consider to re-organize their analysis by first regrouping some of the studies. For example, I would organize the table into the type of study, i.e. case-control studies, GWAS, retrospective versus prospective studies… Maybe make different tables for the different cases. Then, rather of describing each study one by one, I would rather discuss the different studies as an ensemble and try to confront their similarities (eg. DNA methylation, other histone modifications, miRNA…) and their differences (eg. study design, ethnicity, biological sample…), their advantages and inconvenient, what is the most clear/appealing at the moment…
I would advise the authors to reconsider the organization of this review to be accessible to a broader scientific public. Basically, I am asking here to re-write this review to make a better sense out of it, i.e. make in balance the different approaches/types of study, and try to extract potential therapeutic applications, as originally aimed by the authors.
Response: Thanks for your valuable comments. The authors share these argumentations. Unfortunately, reviewing epigenetic markers in such complex disease could be very tricky, given the amount of information, the variability of the available studies on this topic, and the intrinsic variability in patients’ demographics of this pathology. Anyway, we agree with the need of a more critical description of the included studies, of a revision of table 1 and results. According to that, we reorganized table 1 following your and other reviewers’ comments, the results section according to the type of epigenetic marker and biological sample, and the discussion section according to your suggestions. Please find the highlighted corrections in the manuscript with track changes at lines 216-217, 237-246, 256-263, 270-277, 280, 283-289, 297-298, 300-305, 332-333, 378-379 and 420, 424-425 for results section and lines 442-443, 466-487, 496-510, and 516-523 for discussion. Please, also find the revised Table 1, 2 and 3 from page 10 to 19.
Other important points to consider:
- I found their main Table 1 very important to summarize the information, but very difficult to read. I would try to fuse some of the columns to make the table easier to read, i.e. Age/Gender/Ethnicity could be fused; Biological sample and Technique could be fused; Authors could be replaced by the reference number… Anything that could compile some of the information and make the table easier to read.
- In their introduction, the authors mentioned DNA methylation and histone post-translational modifications, without mentioning the functional role of these epigenetic modifications on gene expression. At some point, the authors should clearly indicate that hypermethylation (of the promoter) leads to the repression of the genes, histone/H4 acetylation to activation, H3K27/H3K9 trimethylation to repression… This would help the interpretation of some transcriptional up or down-regulation observed in their case studies.
Response: Thank you for this suggestion. We better described two of the most frequent epigenetic modifications in the introduction and added a description for all the other epigenetic modifications in the discussion. Please find the highlighted corrections in the manuscript with track changes at line 101-108 and 119 (Introduction) and lines 216-217, 237-246, 256-263, 270-277, 280, 283-289, 297-298, 300-305, 332-333, 378-379 and 420, 424-425 for results
- In the results, it is sometime difficult to follow the function of the modifications, see also my previous comment.
For example, the authors should be more clear on how it works. For instance, SOX9 is a disc degeneration related-gene. Does it mean that he has a positive or a negative effect on the pathology? Inhibition of EZH2 ameliorates the progression of IVDD, which means that inhibition of EZH2 leads to a reduced IVDD progression? In this case, it would be more direct to say: SOX9 is a target of EZH2, the writer of the repressive mark H3K27me3. Inhibition of EZH2 leads to the reduction of this repressive mark, and to an up-regulation of the SOX9 target, which could be implicated in slowing down IVDD progression… For instance, pharmaceutical inhibitors of EZH2 actually do exist (used in cancer treatment), and they could maybe be considered here as potential therapeutic treatment for low back pain relief.
To make this review more attractive, this remark could be extended to some other examples described in this paper.
Response: Thank you for your insightful feedback. We addressed a clearer description of the role of the modifications described in this review.
We thank you again for all the interesting suggestions, that hopefully helped us submitting a better written and more relevant paper.
Yours sincerely
Simona Neri, M.D

Reviewer 2 Report
Title
Title is appropriate because it is completely informative about the contents of the paper.
Abstract
The abstract respects the rules of the journal. The background and the aim are interesting. In the design is present the type of study. Methods are better explained. The clinical Impact is present but need to be better explained.
Text
The introduction and the discussion of the study clearly sum up the background of the study. The authors provide a rationale for performing the study on a review of the medical literature. The results are reported clearly and concisely, but to arrange the tables better to show results in the best manner. The paragraph on clinical implications should be improved in the discussion.
References
The reference list follows the format for the journal.
Tables
Tables would illustrate the findings.
Figures
They highlight the key points and they are good quality. In figure 1, the legend have to better explained.
Statistical Analysis
It isn’t needed further checking of data by a statistician reviewer.
General comments
The purpose of the study is original and interesting. The hypothesis is defined. The methods are clear. The study has been structured and carried out correctly about the methodology. Finally, the paragraph on clinical implications should be improved in the discussion.
Author Response
Dear Editor and Reviewers,
First of all, the Authors would like to thank you for the time you dedicated to our manuscript and for your comments and suggestions, which certainly helped us improving the quality of the paper in this second version. It is with great pleasure that we revised our manuscript accordingly. Changes have been made in bold.
Here you can find point-by-point answers to your suggestions and comments. We hope these changes will make the text worth of your consideration.
Best regards,
Simona Neri, M.D.
------------------------------------------------------------------------------------------------------------------------
POINT-BY-POINT ANSWERS:
Reviewer #2:
Title
Title is appropriate because it is completely informative about the contents of the paper.
Abstract
The abstract respects the rules of the journal. The background and the aim are interesting. In the design is present the type of study. Methods are better explained. The clinical Impact is present but need to be better explained.
Response: Thanks for the valuable comment. We tried to better explain the clinical impact of the present review. Please find the highlighted correction in the manuscript with track changes at line 27-28. We also modified the aim at line 119.
Text
The introduction and the discussion of the study clearly sum up the background of the study. The authors provide a rationale for performing the study on a review of the medical literature. The results are reported clearly and concisely, but to arrange the tables better to show results in the best manner. The paragraph on clinical implications should be improved in the discussion.
Response: Thank you for your insightful feedback. As you can see in the revised version, we re-organized table 1 following reviewers’ comments.
Furthermore, clinical implications were better explained in the revised text (lines 446-487, 496-510 and 516-523.
References
The reference list follows the format for the journal.
Tables
Tables would illustrate the findings.
Figures
They highlight the key points and they are good quality. In figure 1, the legend have to better explained.
Response: Thank you for your comment. We revised figure 1 legend. Please find the highlighted correction in the manuscript with track changes at line 67-73.
Statistical Analysis
It isn’t needed further checking of data by a statistician reviewer.
General comments
The purpose of the study is original and interesting. The hypothesis is defined. The methods are clear. The study has been structured and carried out correctly about the methodology. Finally, the paragraph on clinical implications should be improved in the discussion.
Response: Please, see the response above.
--------------------------------------------------------------------------------------------------------------
We thank you again for all the interesting suggestions, that hopefully helped us submitting a better written and more relevant paper.
Yours sincerely
Simona Neri, M.D.

Reviewer 3 Report
This narrative review delineates a comprehensive picture of epigenetic factors related to low back pain and degenerative spine diseases. Table 1 is comprehensive and very easy to read.
1. Figure 1 is not relevant to this article. It is misleading in some way.
2. The author pointed out that the absence of an international consistent low back pain definition is one of the causes that contribute to the heterogeneity of current studies. Other potential causes should be discussed.
3. In conclusion, the author stated "Epigenetics represents a promising field for the identification of factors associated with low back pain." Please address the reasons or evidence supporting this statement in the discussion or result section.
4. In addition, in conclusion, the author stated that more studies are necessary for this field. Please specify what kind of study and why.
Author Response
Dear Editor and Reviewers,
First of all, the Authors would like to thank you for the time you dedicated to our manuscript and for your comments and suggestions, which certainly helped us improving the quality of the paper in this second version. It is with great pleasure that we revised our manuscript accordingly. Changes have been made in bold.
Here you can find point-by-point answers to your suggestions and comments. We hope these changes will make the text worth of your consideration.
Best regards,
Simona Neri, M.D.
------------------------------------------------------------------------------------------------------------------------
Reviewer #3:
This narrative review delineates a comprehensive picture of epigenetic factors related to low back pain and degenerative spine diseases. Table 1 is comprehensive and very easy to read.
- Figure 1 is not relevant to this article. It is misleading in some way.
Response: Thanks for the comment. With figure 1, we tried to provide an immediate and succincte infographic representation of low back pain aetiology, highlighting the epigenetic modulation as one of these. As suggested by reviewer #2, we also added an updated version of figure 1 legend for a better explanation. Please find the highlighted correction in the manuscript with track changes at line 67-73.
- The author pointed out that the absence of an international consistent low back pain definition is one of the causes that contribute to the heterogeneity of current studies. Other potential causes should be discussed.
Response: From lines 409 to 414, we highlighted the limitations of the present review. In terms of research design and demographic, we discovered some variations across the included studies. Another limitation was the lack of a globally recognized definition of low back pain. With this assertion we intended the aforementioned element is a limitation of the review, but it also constitutes a confounding factor reflecting on the heterogeneity of the studies on this topic. We better discussed the potential causes of heterogeneity, please see track changes at line 496-510.
- In conclusion, the author stated, "Epigenetics represents a promising field for the identification of factors associated with low back pain." Please address the reasons or evidence supporting this statement in the discussion or result section.
Response: We addressed the reasons supporting the abovementioned statement. Please find the highlighted correction in the manuscript with track changes at line 537-540.
- In addition, in conclusion, the author stated that more studies are necessary for this field. Please specify what kind of study and why.
Response: We agree, a better definition of further studies needed to be better defined. Please find the highlighted correction in the manuscript with track changes at line 542-543.
--------------------------------------------------------------------------------------------------------------
We thank you again for all the interesting suggestions, that hopefully helped us submitting a better written and more relevant paper.
Yours sincerely
Simona Neri, M.D.

Round 2
Reviewer 1 Report
The authors made an effort to answer my general comments and criticisms of the initial version.
Although I still find this paper largely descriptive, this revised version has largely improved, but there is still place for improvement…
Some minor points to be addressed by the authors before publication:
- Editing aspects: Some of the sentences are too long, sometime complex to understand, and definitely diserve some improvement.
- Actually, authors often refer to « IVD or IVD degeneration »:
For the sake of homogeneity, define once at the beginning « intervertebral discs » (IVD) (actually before the tables where it is extensively used, i.e. from line 54) and use it adequately throughout the text, such as « IVD degeneration… »
And not necessary to use the abbreviation IVDD, it adds some confusion.
- line 164: Table 1 is mentioned, but now it is Table 1, 2, 3... please, describe better.
- From line 245: refer adequately in the text to Table1/vertebral tissues; Table 2/peripheral blood; Table 3/microRNA regulation.
- line 226: it is the description of the first Table:
so replace « Tab 2 » by « Tab 1 ».
- In the Tables, the reference numbers are not the same than in the text: Please double check!
For example: Jiang et al (36) versus (37) in the text...
- line 217: replace « .. and H4 proteins » by « .. and H4 histone proteins »
- At several places in the text: replace « lysin » by « lysine ».
Author Response
Dear Editor and Reviewers,
First of all, the Authors would like to thank you for the time you dedicated to our manuscript and for your comments and suggestions, which certainly helped us improving the quality of the paper in this second version. It is with great pleasure that we revised our manuscript accordingly. Changes have been made in bold.
Here you can find point-by-point answers to your suggestions and comments. We hope these changes will make the text worth of your consideration.
Best regards,
Simona Neri, M.D.
------------------------------------------------------------------------------------------------------------------------
Answer to Reviewer 1:
Thank you for these suggestions, we modified as you requested all the points in our manuscript.
Please find the corrections highlighted in red in the manuscript.
Thanks for your effort in reviewing this paper
--------------------------------------------------------------------------
We thank you again for all the interesting suggestions, that hopefully helped us submitting a better written and more relevant paper.
Yours sincerely
Simona Neri, M.D.